# Quantifying the Resilience Performance of Airport Flight Operation to Severe Weather

**Xinglong Wang** [1], **Ziyan Chen** [1] and **Kenan Li** [1,2,*]

1 College of Air Traffic Management, Civil Aviation University of China, Tianjin 300300, China; xl-wang@cauc.edu.cn (X.W.); 2019031004@cauc.edu.cn (Z.C.)

2 The 28th Research Institute of China Electronics Technology Group Corporation, Nanjing 210007, China

* Correspondence: knli@cauc.edu.cn

**Abstract:** The increased number of severe weather events caused by global warming in recent years is a major turbulence factor for airport operation and results in more irregular flights. Quantifying the system response status towards turbulence is critical, in order for airports to deal with severe weather. For this reason, we propose a resilience framework that is in compliance with resilience theory to evaluate airport flight operations. In this framework, the departure rate (DPR), normal weather baseline (NWB), and nonnegative general resilience (NGR) were defined and used. Meanwhile, the whole process is divided into five phases before and after disturbance, and the system capacities of susceptibility, absorption, adaptation, and recovery are assessed. In order to clarify the performance of the framework towards various severe weather conditions, an analysis was conducted at Beijing Capital Airport in China based on a dataset that includes both the meteorological terminal aviation weather report (METAR) and flight operations from January to July 2021. The results show that the newly proposed resilience framework can commendably reflect airport flight operation performance. The airport flight operation resilience characteristic is different with severe weather. Compared to sandstorms and snow, airport flight operation with stronger robustness was observed during thunderstorm events. The study also confirms that, as the weather warning level increases, the disruption time increases and response time decreases accordingly. The above results could assist researchers and policy makers in clearly understanding the real-world resilience of airport flight operation, in both theory and practice, and responding to emergent disruptive events effectively.

**Keywords:** severe weather event; airport flight operation; resilience assessment

## 1. Introduction

As software and hardware update continuously and because aircraft performance, control technology, ground support, civil aviation laws and regulations, and operating procedures continue to improve, airport flight scheduling is becoming more and more scientific [1]. Therefore, the impact of equipment factors on the safety and efficiency of airport operations is constantly decreasing. Meanwhile, severe weather events, such as typhoons, blizzards, and rainstorms, are occurring more frequently because of global warming [2]. As a result, severe weather has become the main incentive to restrict the normal flight of flights and created huge challenges to the operation air transportation systems. Severe weather events can cause widespread disruptions to the operation of airport infrastructure and even loss of lives, resulting in heavy financial losses. According to the Civil Aviation Administration of China's 2020 Civil Aviation Industry Development Statistical Bulletin, the proportion of abnormal flights due to weather at airports whose throughput accounts for more than 1% of the country has increased from 19.59% to 46.49% from 2014 to 2020 [3].

The impact of severe weather has attracted extensive attention from domestic and international scholars, and much research has been conducted on airport capacity and

delay [4–6]. To better understand the performance of the system during and after severe weather events and maintain the sustainability of the system, the concept of resilience has received more and more attention in the field of transportation and become a new perspective for evaluating the performance of a system in a state of disruption.

In 1973, the concept of resilience was first proposed by the Canadian ecologist Holling, who studied the absorptive capacity of ecosystems to change [7], which was subsequently applied to various fields, such as economics, engineering, and urban management. The Latin word "resiliere" means to bounce back [8]. Gao proposed a multi-dimensional analysis method to determine the resilience of complex networks and found that the three key factors affecting resilience are density, heterogeneity, and symmetry [9].

In recent years, resilience theory has been applied in the field of transportation and shown advantages in dynamic evaluation. The concept of resilience was first introduced to the transportation field by Murray-Tuite [10]. Then, IP and Wang proposed a traffic network structure optimization model to evaluate and analyze the vulnerability and resilience of railway networks [11]. Nogal evaluated the resilience of transportation networks from disruption to recovery under disruptive events and proposed a dynamic equilibrium constraint assignment model [12]. Zhang studied urban traffic resilience based on the spatiotemporal clustering of congestion and found that traffic network resilience conforms to a scale-free distribution [13].

In the aviation field, Gomes studied the concept of vulnerability, as well as the resilience of marine helicopter transportation systems [14]. In 2013, the European Organization for Navigation Safety launched the Enhanced Air Traffic Management (ATM) Resilience Project for emergencies. Subsequently, Gargiulo F proposed a definition of the ATM system's resilience metrics [15]. Palumbo R proposed a new method for the resilience engineering of future ATM systems [16]. Chandramouleeswaran K.R. quantified the resilience of air transport networks using real data [17]. Wang Xinglong introduced the concept of sector network resilience from the perspective of structure and function of system resilience [18–20]. Du et al., proposed a Chinese airport network resilience model and analyzed network performance from the perspectives of structure and dynamic processes [21].

According to previous analyses, most previous studies used topological methods to quantify the resilience of the transportation system [22–24]. The most commonly adopted approach is to simulate an emergency scenario by continuously removing nodes/links in the transportation network, as well as to simulate the gradual dissipation and system recovery scenarios by restoring nodes/links. Every time a node/link is removed or restored, the resilience of the system is quantified. Although these studies provide preliminary explorations for the measurement and evaluation of transportation resilience, they mainly focus on the network level, and few studies quantify the resilience of transportation network nodes or infrastructure, especially the lack of methods to assess the resilience of major transportation hubs. Furthermore, most of them are based on hypothetical emergency scenarios and simulated data, lacking the use of real case data to assess the extent to which airport performance is affected by different factors, which has limitations in reflecting airport flight resilience.

On the above premises, we constructed a conceptual framework of airport flight operation resilience and established a precise evaluation index system. Taking Beijing Capital Airport in China as an example, an airport resilience assessment model was established based on resilience theory. The relationship between the airport flight performance level and resilience was studied from the two dimensions of severe weather type and intensity, and the resilience of the Beijing Capital Airport in China was comprehensively measured. The main contributions of this paper can be summarized as follows:

1. This paper discusses the coupling relationship between severe weather and the resilience of airport flight operations in depth and proposes a resilience metric to quantify the influence of severe weather events on the airport.

2.　This study extended existing resilience research methods to a wider range of severe weather using a variety of severe weather conditions, rather than just one type of severe weather, and drew general and practical conclusions.

3.　The results show that the metric can be used to evaluate the resilience of airport flight operation to severe weather events effectively. Additionally, the research method and approach demonstrated in this paper is transferable to other infrastructure systems.

The rest of paper is organized as follows: the materials and methods are presented in Section 2. Section 3 describes the data and preprocessing conditions. Section 5 introduces the results and discussion. Section 5 also summarizes the entire study.

## 2. Materials and Methods

In the actual operation process, when an airport is affected by special events, such as severe weather, equipment failure, military activities, etc., airport flight operation can effectively respond to risk disturbances and quickly recover to its initial stable state or reach a new state level through adaptive adjustment, thus ensuring the normal operation of the system. To be able to express this process, we propose a quantitative assessment method to determine the airport flight operation resilience capability.

### 2.1. Airport Flight Operation Resilience Capability

Airport flight operation resilience capability is usually measured in terms of the changes in on-time performance, from the perspective of air transport system dynamics [25–27].

This indicator refers to the ratio of the number of flights whose actual departure time is more consistent with the planned departure time to the total number of flights when the air transport department implements the transportation plan, that is, the proportion of non-delayed flights to the total number of flights. This is divided into departure punctuality (DPP) and arrival punctuality (ARP). Taking the DPP as an example, the indicator can be described as:

$$DPP_{T_1,T_2} = \frac{N_{ontime,T_1,T_2}}{N_{schedule,T_1,T_2}} \tag{1}$$

where $N_{schedule,T_1,T_2}$ is the total number of flights that are scheduled to arrive/depart in a given time period $[T_1, T_2)$. Air service delays are defined according to the Civil Aviation Administration of China (CAAC) standard, which considers a flight to be delayed if the difference between its actual departure/arrival time and scheduled departure/arrival time is more than 30 min. $N_{ontime,T_1,T_2}$ is the number of flights in the set whose arrival/departure delay time is less than 30 min. Higher punctuality in a given time indicates the better performance of the airport flight operation system. The value range of punctuality is between 0 and 1.

Although arrival/departure punctuality can generally reflect the performance level of an airport flight operation system, there are some defects. Taking the departure punctuality as an example, the deficiency of this metric is that it can only describe the proportion of the number of flights departing on time in a given time $[T_1, T_2)$ to the total number of scheduled departure flights at that time. DPP does not take either the delayed flights in the time period before $[T_1, T_2)$ or those flights taking off in the time period of $[T_1, T_2)$ into account. It can be seen that the metric has no memory of the flights operation before a given time period $[T_1, T_2)$. Therefore, it is unable to reasonably reflect the actual performance level of the airport flight operation system at a given time. Based on the above analysis, the arrival/departure rate is proposed as a metric with memory of previous flight operation. The definition of departure rate (DPR) is as follows:

$$DPR_{T_1,T_2} = \frac{N_{ontime,T_1,T_2} + N'_{delay,T_1,T_2}}{N_{schedule,T_1,T_2} + N'_{delay,T_1,T_2}} \tag{2}$$

where $N'_{delay,T_1,T_2}$ is the number of flights that have been delayed before time $T_1$ and arrival/departure of those flights in a given time period $[T_1, T_2)$. Compared to punctuality, the arrival/departure rate, which is proposed in this paper, can more accurately describe the actual operation performance of airport flights. The metric not only considers the number of flights that arrive/depart on time in a given time period $[T_1, T_2)$ but also the operation situation of those flights before time period $[T_1, T_2)$.

Ideally, flight performance will always be at the highest level. Correspondingly, the DPR formula shows that the numerator and denominator are equal. That is, there are no delayed or cancelled flights, and all planned flights take off on time. Therefore, the DPR always takes a value of 1. However, in real life, even in normal weather, flight operations will be affected by some uncontrollable factors, such as equipment problems, air traffic control, etc., so the departure DPR will not always be 1 in normal weather. This paper studies the changes in flight performance in severe weather. If the departure rate is 1 in the ideal state, there will be a deviation between the theoretical research and actual situation. Therefore, we define the normal weather baseline (NWD) in this paper, which adopts the flight operation performance in normal weather as the baseline.

### 2.2. Resilience Measurement

Over the past decade, researchers have proposed different approaches to quantify resilience. The "resilience triangle" is a quantitative resilience assessment method proposed by Bruneau in 2003 from a system perspective [28]. This method evaluates the failure process of the system under disturbance events and recovery process after the disturbance by calculating the area of the triangle on the time series, which provides an important reference for the subsequent quantitative evaluation of the system resilience. In 2004, Chang and Shinozuka proposed a probabilistic approach for measuring seismic resilience after earthquake events [29]. In 2008, McDaniels et al., developed a knowledge-based approach using decision flow diagrams, in order to improve our understanding of two dimensions (robustness and rapidity) affecting the resilience of infrastructures [30]. Bueno assessed the degree of socio-ecological resilience from the point of view of system dynamics [31]. This approach, combined with complex network theory, was later applied by Filippini [32]. Most of the existing methods for quantifying resilience lack the assessment of resilience capabilities at all phases. Furthermore, they often overlap with other concepts, such as robustness, vulnerability, and fragility [33]. In addition, several methods mainly focus on the evaluation of performance loss, without considering rapidity and robustness [34]. Cen Nan developed a unifying methodology to assess the resilience of infrastructures in response to various disturbances [35]. This paper adopts a method from the literature [35] to evaluate the resilience of airport flight operations. The system is divided into five stages before and after the disturbance. Figure 1 shows a schematic diagram of the five different stages of system resilience. The vertical axis is the system performance resilience index measurement of resilience performance (MORP). The selection depends on the content of the study. This paper studies the resilience of airport flight operations under severe weather conditions and uses the DPP and DRR as the indicators of airport flight performance resilience. The value range of the MORP value fluctuates between [0,1], where 0 is the system is in a paralyzed state and 1 is the performance of the system in an ideal state.

Figure 1 shows the five stages of MORP under external interference. Among them, $t_d'$ time indicates the beginning of a severe weather event. $t_d$ time indicates the moment when the system performance lever starts to degrade. $t_r$ represents the moment when MORP drops to the point. MORP reaches a new stable stage at time $t_{ns}$. Different indicators were defined for each stage, and the specific indicators of this resilience assessment model are as follows:

1. The first phase is the initial stable phase. The time period $t_0 < t < t_d'$ is the initial stable stage of the system, when the system is not disturbed by the outside world, and the system performance level is the initial performance value.

2.  The second phase is the response phase. The time period $t_d' < t < t_d$ is the response stage of the system under external interference, during which the system has been disturbed, but the system performance level still remains at the initial performance value. During this phase, the system's susceptibility capability can be assessed by identifying appropriate measures. The selection of the appropriate MORP depends on the specific service provided by the system under analysis. It is assumed that disruptive events occur at $t_d'$ and the MORP value drops at $t_d$. It should be noted that in many cases, $t_d'$ might not be equal to $t_d$, and the $t_d - t_d'$ delay depends on the selection of the MORP and disruptive event. For instance, it could take several hours for passengers to lose air services due to general severe weather events, while it might only take seconds for same passengers to lose services due to natural hazards, such as earthquakes. System sensitivity could be used to characterize the performance of this stage. System susceptibility is defined as "the inability of a system to avoid being hit by a threat mechanism" [36].

3.  The third phase is the disruptive phase (DP), in which the system performance starts dropping at time $t_d$, until the lowest level is reached at time $t_r$. During this phase, the system absorptive capability can be assessed by identifying appropriate measures.

4.  The fourth phase is the recovery phase, in which the system performance increases until the new steady level is achieved. During this phase $t_r < t < t_{ns}$, the adaptive and restorative capabilities of the system can be assessed by developing appropriate measures.

5.  The fifth phase is the new stable phase, in which system performance reaches and maintains a new steady level. It should be noted that the new stable level may be equal to, lower than, or even higher than the initial level. During this phase, the system recovery capability can be assessed by identifying appropriate measures.

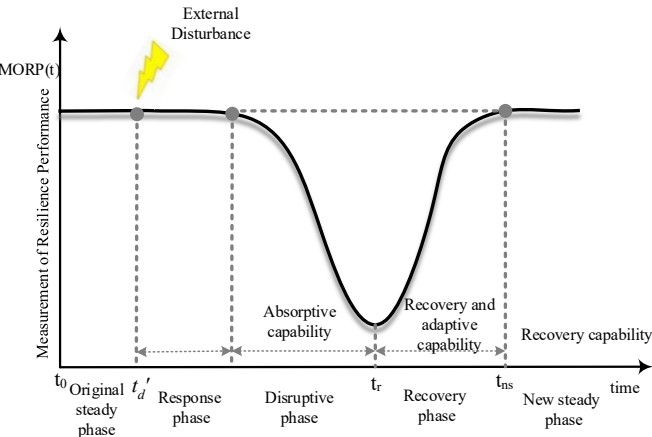

**Figure 1.** Schematic diagram of system resilience.

**Definition 1:** Response time (RST) is used to indicate the susceptibility of the system. It can be represented with Formula (3):

$$RST = t_d - t_d' \tag{3}$$

**Definition 2:** Disruption time (DST) represents the period of time between when system performance begins to drop at time $t_d$, as well as the time with the worst system performance $t_r$. It can be represented with Formula (4):

$$DST = t_r - t_d \tag{4}$$

**Definition 3:** Rapidity in the disruptive phase (RAPI$_{DP}$) is used to measure the speed of system performance degradation. It can be represented with Formula (5):

$$RAPI_{DP} = \frac{MORP(t_d) - MORP(t_r)}{t_r - t_d} \tag{5}$$

$MORP(t_d)$ is the system resilience capability value corresponding to time $t_d$, when the performance begins to drop. $MORP(t_r)$ is the system resilience capability value corresponding to the lowest performance time $t_r$.

**Definition 4:** Robustness (R) quantifies the minimum value of MORP between $t_d$ and $t_{ns}$. This measure is able to identify the maximum impact of disruptive events. It can be calculated by following Formula (6):

$$R = min\{MORP(t)\}(t_d < t < t_{ns}) \tag{6}$$

**Definition 5:** Recovery time (RCT) represents the period of time between the worst system performance at time $t_r$ and new steady level at time $t_{ns}$. It can be represented with Formula (7):

$$RCT = t_{ns} - t_r \tag{7}$$

**Definition 6:** Rapidity in the recovery phase (RAPI$_{RP}$) is used to measure the speed of system resilience capability of recovery in the recovery phase. It can be represented with Formula (8):

$$RAPI_{RP} = \frac{MORP(t_{ns}) - MORP(t_r)}{t_{ns} - t_r} \tag{8}$$

$MORP(t_{ns})$ is the system resilience capability value corresponding to time $t_{ns}$, when the system performance reaches a new stable stage.

**Definition 7:** Recover ability (RA) is used to measure the size between the resilience capability achieved by the system in the new stable phase and resilience capability of the initial stable phase. It can be expressed with Equation (9):

$$RA = \left| \frac{MORP(t_{ns}) - MORP(t_r)}{MORP(t_d) - MORP(t_r)} \right| \tag{9}$$

**Definition 8:** Loss of performance (LOP) is used to measure the performance degradation of the system caused by the negative impact during the whole process of the disturbance event, which can be expressed with Equation (10):

$$LOP = \int_{t_d}^{t_{ns}} [MORP_{week}(t) - MORP(t)]dt \tag{10}$$

$MORP_{week}(t)$ represents the mean MORP in the day of week as the baseline of the system resilience capability. The calculation method for $MORP_{week}(t)$ should be based on the actual data collected from research objects.

**Definition 9:** Time-averaged performance loss (TAPL) is used to measure the system performance degradation caused by negative impact during the entire process of the disturbance event, which can be expressed with Equation (11).

$$TAPL = \frac{\int_{t_d}^{t_{ns}} [MORP_{week}(t) - MORP(t)]dt}{t_{ns} - t_d} \tag{11}$$

The above provides a comprehensive introduction to the five stages of the resilience process. Different metrics are proposed for assessing the system's susceptibility capacity,

absorptive capacity, adaptive capacity, and recover capability. For ease of understanding, Table 1 summarizes the different resilience phases.

**Table 1.** Summary of different resilience phases.

| Phases | Time Scope | Capabilities | Measurements |
|---|---|---|---|
| Original steady phase | $t_0 < t < t_d'$ | - | - |
| Response phase | $t_d' < t < t_d$ | Susceptibility | RST |
| Disruptive phase | $t_d < t < t_r$ | Absorptive capability | R, RAPI$_{DP}$, DSS |
| Recovery phase | $t_r < t < t_{ns}$ | Recovery and adaptive capability | RCT, RAPI$_{RP}$ |
| New steady phase | $t \geq t_{ns}$ | Recovery capability | RA |

In order to comprehensively compare and analyze the resilience of different systems under different disturbance events from an overall perspective, Cen Nan proposed the general resilience (GR) index, which is used to comprehensively measure the entire resilience process of the system, both before and after the external disturbance occurs [30]. The definition is as follows:

$$
\begin{aligned}
GR &= f(R, RAPI_{DP}, RAPI_{RP}, TAPL, RA) \\
&= R \times \left( \frac{RAPI_{RP}}{RAPI_{DP}} \right) \times \left( TAPL^{-1} \right) \times RA
\end{aligned}
\tag{12}
$$

The disadvantage of this index is that it does not take into account that the system performance will recover again after reaching the lowest value of 0. Therefore, if the robustness under a certain interference event is 0, then it is unreasonable to directly consider that the comprehensive resilience value of the system under this event is 0. In addition, the occurrence of this situation can also lead to the inability to analyze and compare comprehensive resilience values between different systems. Therefore, on this basis, this paper proposes a new comprehensive resilience index, i.e., the nonnegative general resilience (NGR), which limits the robustness of the system and converts it to a non-negative value. It is shown in the following Formula (13):

$$
\begin{aligned}
NGR &= f(R, RAPI_{DP}, RAPI_{RP}, TAPL, RA) \\
&= e^R \times \left( \frac{RAPI_{RP}}{RAPI_{DP}} \right) \times \left( TAPL^{-1} \right) \times R
\end{aligned}
\tag{13}
$$

The improved indicators can effectively avoid the above problems and evaluate the characteristics and behavior changes in airport flight operations under different severe weather disturbances from an overall perspective, thus reflecting the dynamic resistance, adaptation, absorption, and final recovery of airport flight operations to disturbance event characteristics. This paper focuses on resilience quantification after the appearance of the negative effects, so susceptibility is not considered. Therefore, NGR does not include the indicator RST, which reflects susceptibility.

If the system is more capable of resisting a disruptive event or force (large R, small RAPIDP), it will be more resilient (large NGR); if the system is more capable of reducing the magnitude and duration of the deviation in its performance level between the original state and new steady state (small TAPL, large RAPI$_{RP}$), then it will be more resilient (large NGR). Additionally, NGR incorporates the possibility of improving the system performance after the occurrence of the disruptive event. If the new performance level is higher than the original (large RA), the system is more resilient (large NGR).

The value of NGR equals 0 in the following relevant cases:

1. System performance immediately drops to its lowest level under the disturbance ($RAPI_{DP} \to \infty$, i.e., no absorptive capability);
2. System performance never increases past the lower level, R, which is the new steady phase ($RAPI_{RP} = 0$, i.e., no adaptive and restorative capability).

## 3. Data Description and Preprocessing

### 3.1. Data Description

In this study, Beijing Capital Airport in China was selected as the study site. The data set used in this study includes three parts: (1) flight departure data (486,675 pieces) were provided by the Beijing Capital Airport in China during the period from 00:00 on 1 January 2021 to 00:00 on 1 August 2021 (a total of 5088 h, including the departure airport, arrival airport, planned departure time, actual departure time, and whether the flight is cancelled); (2) the meteorological terminal aviation weather report (METAR), which was recorded every 30 min during the same period and obtained from the OGIMET webpage on 20 September 2021 (www.ogimet.com, accessed on 20 September 2021)—the data set includes a total of 10,176 message data (including the elements of wind, temperature, dew temperature, visibility, weather, cloud, pressure, etc.); (3) the weather warning information for Shunyi district, Beijing, issued by the Beijing Meteorological Observatory (including the dangerous weather information of blue, yellow, orange, and red levels).

### 3.2. Data Preprocessing

This paper studies the flight performance of Beijing Capital Airport. The flight data contained all of the arrival and departure flight information from 00:00 on 1 January 2021 to 00:00 on 1 August 2021. However, landing flights were affected by severe weather at the studied airport, as well as by severe weather at the departure airport and en route. Arrival flight information is not completely accurate for studying flight operations at a specific airport, and there are many uncontrollable factors that would reduce the credibility of the research results. Therefore, this paper only uses the departure flight operation data. There is incomplete data in the airport flight operation data; for example, the planned departure time or actual departure time data information are missing in a certain flight information. These flight data were deleted. In addition, some airport flight operation data may seem complete but not make sense—for example, if the actual departure time was the same as the actual arrival time, and if the planned departure airport was the same as the planned arrival airport. These data were also deleted.

This paper filters out the time periods in which there was normal and severe weather, based on METAR messages. When the term "CAVOK" appears on a day when the wind speed is less than 5 m/s, it is considered to be normal weather that has no impact on the flight operation of the airport. First, this paper filters out the message information of these normal weather events and records the corresponding time information. In addition, if one day had normal weather but the previous day had bad weather, this may affect the flight operation under the current normal weather. In order to avoid the occurrence of the above situation, this paper further filters out the message information regarding normal weather for three consecutive days or more and uses the data from the third day and later as normal weather data. Finally, according to some coding identifiers, i.e., TS (thunderstorm), SN (snow), SA (sand), and RA (rain), the time period information of different types of severe weather is screened out.

According to the regulations of the Civil Aviation Administration of China, the whole year is divided into two flight schedule seasons, i.e., the summer–autumn and winter–autumn seasons, and the regular flight plans that usually occupy more than 70% of all flight plans are the same in each season and different between seasons. This first season is implemented from the last Sunday in March to the last Saturday in October, and the second is from the last Sunday in October to the last Saturday in March of the following year. The flight plan runs on a weekly basis, and the normal weather during the flight plan time is screened and classified from Monday to Sunday. Since there are few cases of normal weather for a continuous week, this paper uses a splicing method to form a week of normal weather. For example, the flight operation baseline under Monday in the summer–autumn flight schedule is calculated by summing and averaging the flight operation performance of all of the Monday times in normal weather conditions.

The METAR is recorded every 30 min, and this paper conducts research based on the data information in the METAR. Therefore, in order to reasonably explore the influence of severe weather on airport flight operations, this paper divides the airport traffic flow in a day into 48 periods, according to periods of 30 min.

## 4. Results and Discussion

After previous data preprocessing and other operations, a total of seven severe weather events were screened out, including two snowfall events (SN1 on 18 January and SN2 on 25 January), one sandstorm event (SD on 14 March), and four thunderstorm events (TS1 on 1 July, TS2 on 5 July, TS3 on 11 July, and TS4 on 26 July). According to weather warning information, there was no weather warning information during SN1, SN2, and SD. During TS1, there was a blue warning; during TS2, a yellow warning was issued; during TS3, an orange warning was issued; and a red warning message was issued during TS4. The intensity of severe weather events is classified according to the level of weather warning information, which means that TS4 was the most serious.

### 4.1. Departure Punctuality and Departure Rate

In order to effectively evaluate the effectiveness of the proposed DPR, this paper uses the DPP and DPR to calculate the flight operation data under seven severe weather events, according to a 30-min time period. It can be seen from Figure 2 that DPR is always above DPP under all severe weathers. The change trend of both is roughly the same, but DPP fluctuates greatly. However, the change trends of TS1 and TS2 in Figure 2 are obviously different. It can be seen from Figure 2d that the change trends in DPP and DPR were basically the same before 16:30, and they both decreased to 0 quickly after being affected by severe weather. This shows that the airport operation performance absorbs the impact of severe weather events quickly; the airport flight operation was seriously affected, and the performance dropped sharply. After that, DPP remains at 0 for 5 units of time and then rose rapidly, while the change in DPR over this period showed a gradual and volatile rise. A similar regularity is also shown in Figure 2e. Before 12:30, the change trends of DPP and DPR were basically the same; after that, DPR still showed a fluctuating and slow upward trend under this severe weather event; however, DPP had a long period of time (20 units of time) value that remains at 0. This shows that the airport was severely affected, and that the airport operation performance was almost paralyzed at this time. After that, the DPP rose sharply to 1 in the next 3 units of time, which indicates that the airport operation performance recovered to a new stable phase. According to METAR and weather warning information, it can be seen that TS2 was a moderate-intensity thunderstorm event.

It is obvious that it is not reasonable to use DPP as an indicator of airport flight performance resilience. Firstly, it is unlikely that a 20-unit time (10-h) airport shutdown will occur due to a thunderstorm with moderate rain. Secondly, if there is long-term airport paralysis, this will result in a backlog of flights that have not been able to fly during this time. According to the actual coordinated operation of airport flights, it is not difficult to find that, in this case, it is difficult for the airport flight operation system to coordinate to release all of the previously backlogged flights within one hour. In contrast, since DPR takes the fact that the previously delayed flights take off later on into account, using DPR as the resilience index of airport flight performance can show the real changes in the entire process of airport operating performance under severe weather events more appropriately.

In summary, DPP only considers whether each flight takes off on time, and the DPR proposed in this paper not only considers whether the flight takes off on time but also the situation of taking off after the delayed flight, which is closer to the actual situation than the on-time rate. However, DPP places less emphasis on flight restoration. In the next step, airport flight restoration can be achieved through a flight sequencing model, which can better reflect the restoration of airport flight performance.

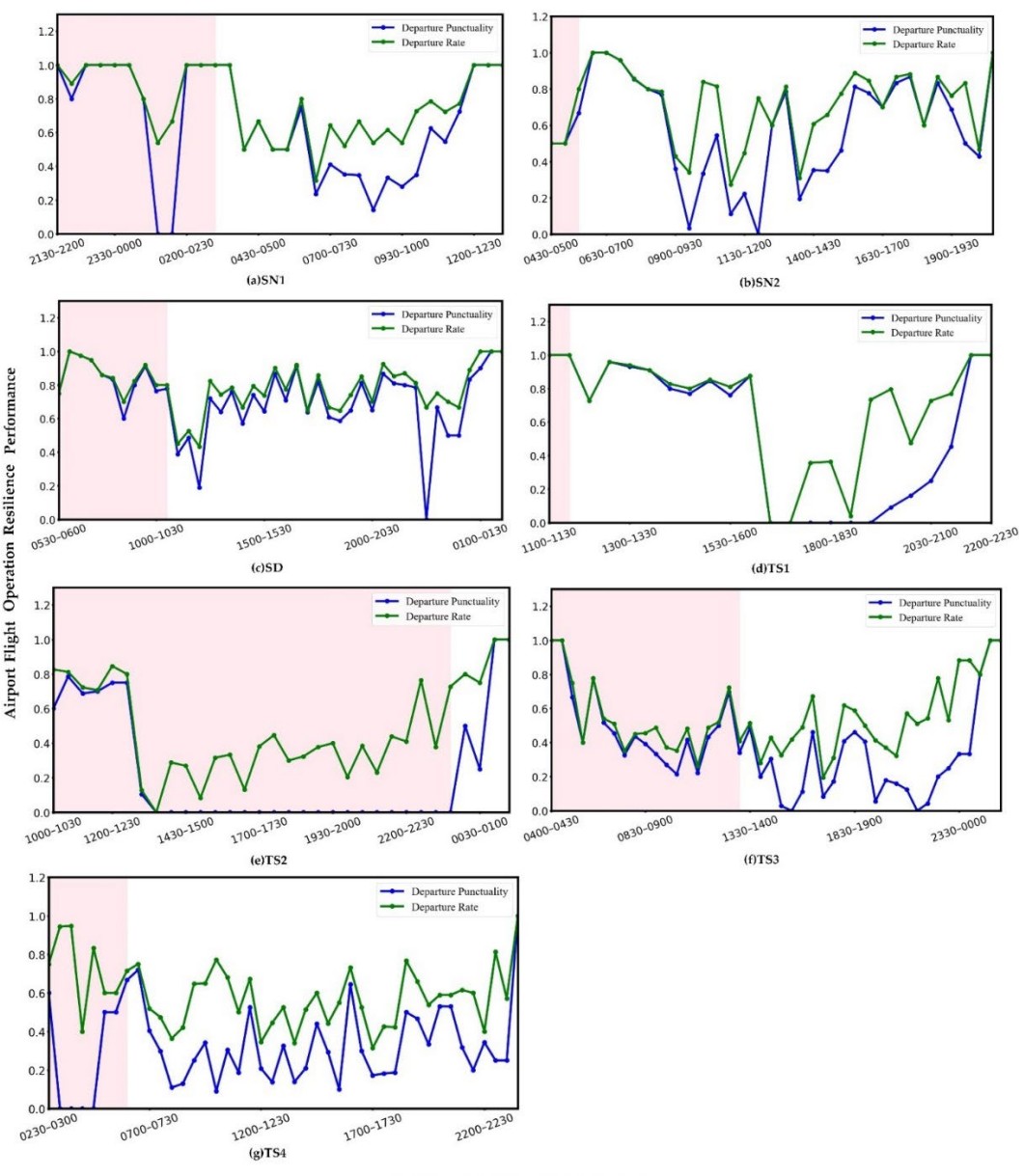

**Figure 2.** Comparison of DPR and DPP under severe weather conditions. The red area represents the occurrence period of severe weather, the green line segment represents DPR, and the blue line segment represents DPP.

### 4.2. Resilience Assessment of Airport Flight Operations under Severe Weather

Figures 3–5 depict the variation in the DPR under snow, sandstorm, and thunderstorm events, respectively. The black line represents the ideal baseline of the airport flight operation, and the ideal baseline indicates that the airport flight operation performance always maintains the highest level under ideal conditions. The airport flight operation performance level is maintained at 1. The yellow line reflects the flight performance of the airport under normal weather. It can be seen that the yellow line fluctuates slightly, indicating that, even if the current weather is normal, there may be flight delays or cancellations. The green line represents the change in the flight performance of the airport in severe weather. Compared to the yellow curve of the system performance in normal weather conditions, there is a certain decline and recovery. The area of the part shaded in grey is the loss of resilience performance in airport flight operation. The four dashed lines represent the time

when severe weather begins ($t_d^{'}$), the airport flight performance begins to decline ($t_d$), the performance reaches its lowest point ($t_r$), and the performance returns to a stable state ($t_{ns}$).

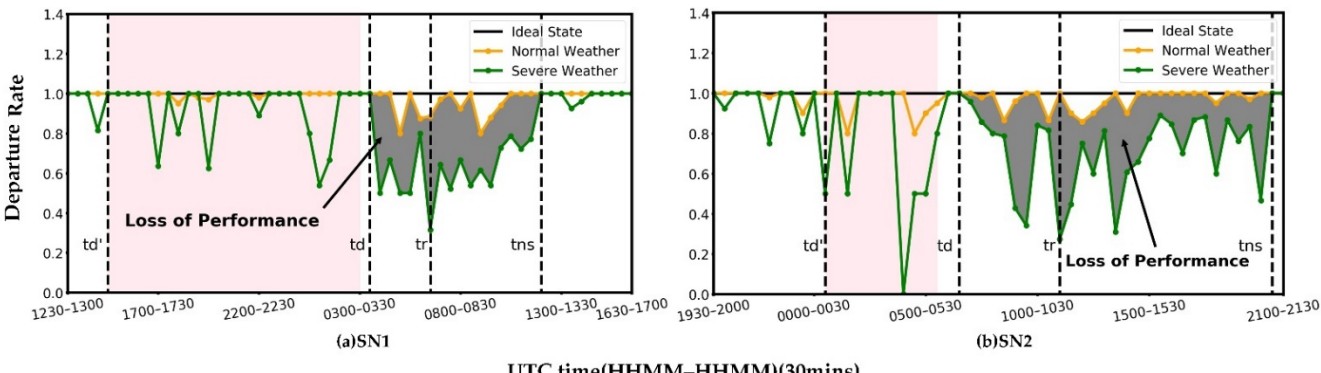

**Figure 3.** Airport flight operation resilience performance under SNs.

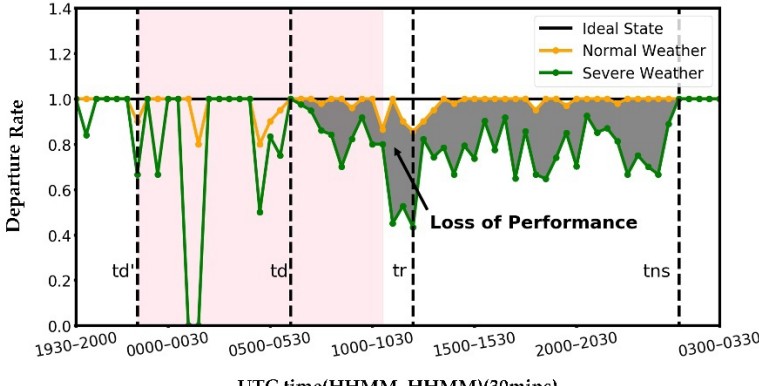

**Figure 4.** Airport flight operation resilience performance under SD.

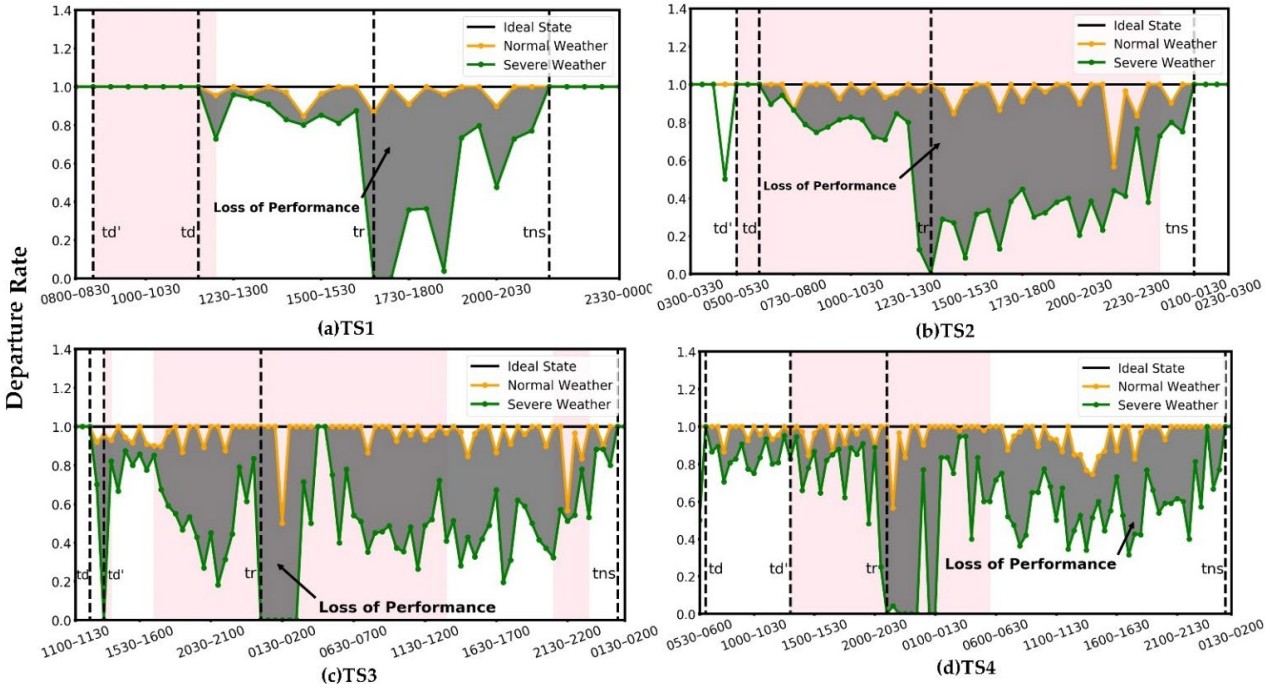

**Figure 5.** Airport flight operation resilience performance under TSs.

In actual operation, it was found that these two key time nodes are easy to identify when the severe weather starts and performance reaches the lowest point. However, due to the large fluctuations in the departure rate, it is difficult to identify the time when the performance begins to decline and performance returns to a stable state. In this paper, the time when the performance begins to decline is identified as follows: after the time, the airport flight operation performance experiences a significant degradation in the duration of performance, rather than a short decline and then a rapid rise (the duration should be greater than six units of time). Similarly, the identification rule for the time when the performance recovers to a stable time also requires a duration greater than six units of time.

Figure 3 shows the change process of airport flight operation performance under two snowy weather events. It can be seen that, in the response phase, the DPR increases rapidly after a slight decrease. The lowest point of MORP was between 0.2 and 0.4 during the whole change process.

Figure 4 shows the change process of airport flight operation performance under sandstorm conditions. It can be seen that, in the response phase, the DPR rose rapidly after being 0 because there was only one scheduled flight in this time period and it did not take off on time. The lowest point of MORP in the whole process is between 0.4 and 0.5.

Figure 5 shows the change process of airport flight operation performance under thunderstorm conditions. It can be seen that, under thunderstorm events, MORP reaches its lowest of value 0.

In all severe weather events, as the severe weather dissipates, the performance level gradually returns to the original level, and the recovery capability index RA of the airport flight performance is 1. When compared to each other, the airport flight performance is not as degraded when sandstorms and snowfall events occur, showing good robustness.

In addition, it can be seen from Figures 3–5 that there is a certain difference between the ideal baseline and NWB. The comparison of LOP under different baseline conditions is shown in Table 2. The experimental results show that, from the perspective of flight performance loss value, the performance loss calculated by taking the two baseline conditions under severe weather is not much different (9.09%), while in severe weather, due to the long duration, the difference is huge (reach to 17.56%). It shows that the performance loss value calculated with the ideal baseline becomes more and more distant from the actual situation as the duration of severe weather increases. Taking NWB, the impact of normalization has been fully considered, according to the actual situation of the airport, and it has a certain robustness, so the calculated performance loss is more realistic and convincing.

**Table 2.** Comparison of LOP in two states.

| Severe Weather Events | LOP of Ideal State | LOP of Normal Weather | Difference | Percentage Error |
|:---:|:---:|:---:|:---:|:---:|
| SN 1 | 6.1872 | 5.2629 | 0.9243 | 17.56% |
| SN 2 | 8.2305 | 7.3232 | 0.9073 | 12.38% |
| SD | 8.5345 | 7.8235 | 0.7110 | 9.09% |
| TS 1 | 7.0419 | 6.3723 | 0.6696 | 10.51% |
| TS 2 | 17.5957 | 15.8681 | 1.7276 | 10.89% |
| TS 3 | 33.9503 | 30.7412 | 3.2091 | 10.44% |
| TS 4 | 32.5233 | 28.6823 | 3.8410 | 13.39% |

For thunderstorm events, the lowest airport flight performance value was 0, and the robustness was poor. Detailed information can be seen in Table 3. In all severe weather events, the RCT of the airport flight performance level is generally longer than the DST, that is, roughly 1 to 2 times longer. The short−term damage and long−term recovery phenomena cause the $RAPI_{RP}$ to be generally lower than the $RAPI_{DP}$, and the $RAPI_{DP}$ is 1 to 2 times that of the $RAPI_{RP}$.

**Table 3.** Summary of calculation results under different severe weather conditions.

| Severe Weather Events | RST (30 mins) | DST (30 mins) | RCT (30 mins) | RAPI$_{DP}$ | RAPI$_{RP}$ |
|---|---|---|---|---|---|
| SN 1 | 26 | 6 | 11 | 0.1140 | 0.0622 |
| SN 2 | 12 | 9 | 19 | 0.0808 | 0.0383 |
| SD | 15 | 12 | 26 | 0.0472 | 0.0218 |
| TS 1 | 4 | 10 | 10 | 0.1 | 0.1 |
| TS 2 | 2 | 15 | 23 | 0.0667 | 0.0435 |
| TS 3 | −2 | 24 | 50 | 0.0417 | 0.0200 |
| TS 4 | −14 | 30 | 56 | 0.0333 | 0.0179 |

It can be seen that, under thunderstorm events, as the warning level increases, the RST decreases and DST increases accordingly (Figure 6). In addition, the RSTs of SN1, SN2, SD, TS1, and TS2 were all positive values; the RSTs of SN1, SN2, and SD were relatively long; and the RSTs of TS3 and TS4 were negative values. This may be related to the intensity and type of severe weather, as well as when it occurred. For example, in SN1 and SN2, airport flight operations were not immediately affected. With the passage of time, the temperature at night decreased, and snow and ice gradually appear on the ground and surface of the aircraft, which has a great impact on the flight operation of the airport. It can be seen that the impact of snowfall events on airport flight operations has a lagging effect. Under TS3 and TS4, the RST was negative, which means that the flight performance of the airport had been reduced before the thunderstorm starts. An orange weather warning was issued during the occurrence of TS3, and a red weather warning was issued during the occurrence of TS4. It can be seen that serious thunderstorm events have a leading effect on the airport flight operation system. As seen from the figure, the RSTs were −2 and −14 under TS3 and TS4, respectively. We found that this was due to high winds, low clouds, and lightning at the airport prior to the thunderstorm, which affected passenger travel activities. At the same time, the relevant management personnel adjusted the corresponding flight plan according to the weather information. Therefore, before the occurrence of severe weather, the operational resilience performance of airport flights has a slight downward trend, compared to the baseline under normal weather conditions. In general, thunderstorms affect other places before they affect airports, meaning that flight scheduling and passenger travel are already affected before the onset of a thunderstorm.

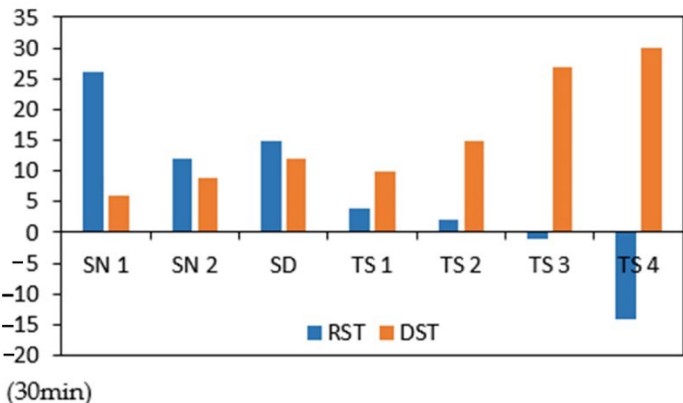

**Figure 6.** Response time and disruption time under severe weather conditions.

In order to analyze and compare GR and NGR, this paper calculates the values of GR and NGR under the corresponding severe weather events, as shown in Figure 7. The GRs under events TS1 to TS4 were all 0. According to Formula (12), this was due to the fact that the airport flight performance level decreased to its lowest value of 0 at a certain moment; that is to say, the robustness R was 0. The value of GR was 0, which indicates that the airport flight operation system had no "rebound" process under the interference of TS1

to TS4. According to Figure 5, it can be seen that, under the interference of TS1 to TS4, the performance of the airport flight operation system was only 0 in a certain period of time. After that, as the severe weather dissipated and coordinated scheduling was managed, the system gradually returned to its original state. In addition, since the GRs of TS4 to TS7 were all 0, the regularity between these severe weather events and GR cannot be explored. This shows that the GR index is unreasonable for carrying out comprehensive assessments on the resilience of airport flight operations.

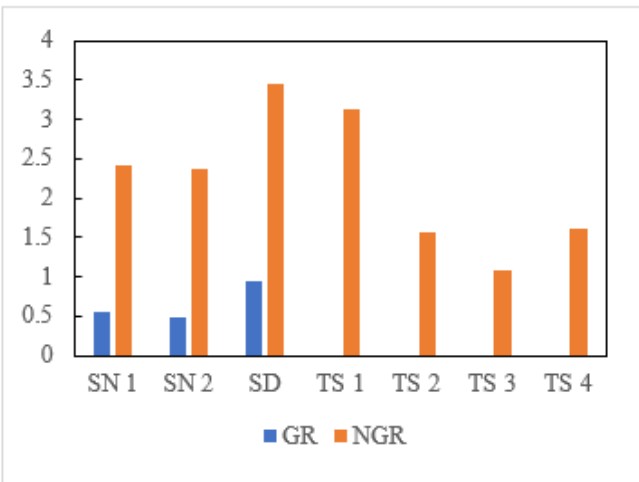

**Figure 7.** Comparison of GR and NGR.

The NGR proposed in this paper avoids such problems. Airport flight operation under SD disturbance have the largest NGR, indicating that the airport's flight operation performance level can return to its initial steady state relatively quickly. Under the interference of SN1 and SN2, the NGR performance was relatively balanced. Under the events from TS1 to TS4, as the weather warning level increased, the NGR first decreased and then increased slightly. Using the weather warning level to characterize the intensity of severe weather, the NGR could reflect the intensity of severe weather. The NGR probably exists in the recovery process of transportation resilience.

In this paper, we measured the resilience behavior of an already-built airport under several severe weather events. Further resilience analyses should be carried out to determine the influence of infrastructure changes to airport operation, for example, the influence of a CAT II runway coming into use at ZPPP in 2015. It could serve as a reference for airport design and planning.

## 5. Conclusions

In this paper, based on the flight operation data, METAR messages, and weather warning data from Beijing Capital Airport from 1 January 2021, to 1 August 2021, we established a framework to determine airport flight operation resilience under severe weather conditions. We generalized existing resilience research methods to a wider range of severe weather scenarios, so that more general and practical conclusions could be drawn. The main findings can be summarized as follows.

First, the concept of airport flight operation resilience was proposed, and DPR is proposed for the problem that delayed flights will still take off in the following time period. The results suggest that, compared to DPP, DPR can better reflect the actual performance of airport flight operations.

Second, considering that the airport is also affected by various other uncontrollable factors during normal weather conditions, flight performance will not always be at the highest level. Therefore, this paper takes airport flight performance level under normal weather conditions as the baseline. The results show that, compared to the ideal state,

the error of LOP calculated with the flight performance in normal weather as the baseline is smaller.

Third, in a comprehensive assessment of the overall resilience of the system, GR does not take into account that when the robustness is 0, the system performance may increase throughout the recovery process. The NGR proposed in this paper can effectively avoid this situation.

Finally, we also found that the robustness of the system is lower under thunderstorm events, compared to snowfall and sandstorm events. Under thunderstorm events, with the increase in the weather warning level, DST increases and RST decreases accordingly. Snowfall has a lagging effect on the impact of the airport system, and severe thunderstorm events have a leading effect on the impact of the airport system.

The analysis of the resilience assessment can obtain the overall impact of different types, intensities, and durations of weather on airport flight operations. The next step is to quantify its impact through in-depth learning. Future weather conditions can be obtained through TAF, for the purposes of predicting airport flight operation conditions, in order to aid decision makers.

**Author Contributions:** Conceptualization, X.W., Z.C. and K.L.; methodology, X.W., Z.C. and K.L.; validation, X.W., Z.C. and K.L.; formal analysis, X.W., Z.C. and K.L.; investigation, X.W., Z.C. and K.L.; Writing—Original draft preparation, X.W., Z.C. and K.L.; Writing— Review and editing, X.W. and K.L.; visualization, Z.C.; supervision, X.W. and K.L. All authors have read and agreed to the published version of the manuscript.

**Funding:** This research was funded by National Natural Science Foundation of China (No. 62173332), Fundamental Research Funds for the Central Universities (No. 3122015C023, No. 3122019191) and State Key Laboratory of Air Traffic Management System and Technology (No. SKLATM202009).

**Institutional Review Board Statement:** Not applicable.

**Informed Consent Statement:** Not applicable.

**Data Availability Statement:** The data presented in this study are available on request from the corresponding author. The data are not publicly available due to a research issue.

**Conflicts of Interest:** The authors declare no conflict of interest.

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
