# Peer review of "Quantifying the Resilience Performance of Airport Flight Operation to Severe Weather"

_aerospace, doi:10.3390/aerospace9070344_

Round 1

Reviewer 1 Report

The authors present a new way to measure the resilience of an airport's flight operations during severe weather events. The article reviews the antecedents in the measure of resilience of airports during adverse meteorological phenomena, carrying out a good quality bibliographical review. Based on the shortcomings of the resilience measures used up to now, two original modifications are proposed that allow for a more useful resilience measure (it is modified so that it does not reach the null value of robustness and incorporate delayed flights that take off in a certain data interval). This allows obtaining a more consistent measure of resilience than the previous ones.

The resilience measure applies to various severe weather events at Beijing Capital Airport. It exposes how the measure of resilience evolves in these cases. It also compares the correlation between resilience indicators and METAR weather advisories.

In addition to previous original contributions, the article has a good presentation quality. However, the article is not in a state that allows its publication. In my opinion, the following points should be improved so that the quality of the article rises:

1-       Correct writing errors, figure numbering, punctuation … Important is the error in equation (12), where R, the Robustness, is missing (ref [27]). Without this correction, subsequent comments are not understood.

2-       It is necessary to do a general review of the writing and grammar in English (Moderate English changes required)

3-       A discussion on how this resilience measure can be used in airport design and planning is missing. Also for the comparison of some infrastructures with others. What is seen in the article helps us to measure the behavior of an airport already built. How can these measurements be used for the design of new ones or compare with each other?

Reviewer 2 Report

Summary:

This paper proposes an approach to characterize airport resilience to capacity decreasing events using performance metrics, extended from previous work, to specifically account for aviation-related behaviors and phenomena. The authors provide a thorough literature review and a detailed description of the metrics proposed including the motivation for any modifications relative to previous work. The authors further describe the data processing steps involved, including the rationale for excluding certain data. The authors present compelling results that are well illustrated and are coupled with insights regarding the operational impacts associated.

Review:

This paper addresses an important research topic that is highly relevant and topical. While the research is more than adequate to publish, the paper itself suffers from technical writing flaws that cannot be overlooked as they significantly impact both readability and comprehension. As such, it is the opinion of this reviewer that the paper be revised, ideally with the assistance of a native English speaker experienced in technical writing, before publication.

Comments:

  1. As mentioned, the grammatical and technical writing flaws are numerous and beyond the reviewer's ability to comprehensively catalog, but to list a few.
    1. There is a spelling and grammatical error in the title: Quantity should be Quantify (noun), but more accurately it should be Quantifying (verb).
    2. Lines 188-190 "The vertical axis is the system performance resilience index measurement of performance resilience (MORP) that  changes with time, which represents the system performance resilience index."  This is a circular sentence that defines a term using the term itself.
    3. Lines 286-293 "The disadvantage of this index is that when the system is in the process of external disturbance events, the system performance level reaches the lowest value of 0, that is, the robustness of the system is 0, and the calculated comprehensive resilience index value is 0. At this time, from the overall point of view, it means that the comprehensive resilience of the system is 0. This indicator does not take into account that the system performance will recover again after reaching the lowest value of 0. Therefore, if the robustness is 0 under a certain interference event, it is obviously unreasonable to directly consider that the comprehensive resilience value of the system under this event is 0." This paragraph is so repetitive that it obscures the message of the authors.
    4. The use of the terms disaster, destruction, etc. are not appropriate nor are they the terms of art in this domain.
  2. The DPR metric is a key contribution of this work as it incorporates the backlog of demand due to delays. In the development of this metric, the authors leverage the previous time-period's delayed demand, which they further demonstrate as having the appropriate impact in the resilience metrics. The comment here is whether the authors have considered the use of a simple queuing model or other simulation to better reflect the demand propagation under delays, potentially spanning multiple time periods. The request here is for justification or discussion as opposed to a need to actually conduct the work in this regard.
  3. In the final paragraph of the paper, the authors state that "In addition, the research results are helpful for aviation decision makers to take measures to reduce the impact of severe weather in accordance with the changes in airport flight performance under severe weather events, which is of great significance for reducing flight delays and ensuring the safety of airport flight operation systems. " To do so would require that the metrics provide actionable insights under forecasted, as opposed to observed, data. While such decision support is the ultimate objective, it would be valuable for the authors to discuss the extensibility of the model for leveraging forecast data, the challenges entailed with validating the predictability of the resulting metrics, and how the metrics themselves could offer insight to decision makers regarding which measures should be applied when to reduce the impact of severe weather.
